

# Identifying app components that promote physical activity: a group concept mapping study

Maya Braun[1], Stéphanie Carlier[2], Femke De Backere[2], Marie Van De Velde[1], Filip De Turck[2], Geert Crombez[1] and Annick L. De Paepe[1]

[1] Experimental Clinical and Health Psychology, Universiteit Gent, Ghent, Belgium
[2] IDLab, Department of Information Technology - imec, Universiteit Gent, Ghent, Belgium

## ABSTRACT

**Background**. Digital interventions are a promising avenue to promote physical activity in healthy adults. Current practices recommend to include end-users early on in the development process. This study focuses on the wishes and needs of users regarding an a mobile health (mHealth) application that promotes physical activity in healthy adults, and on the differences between participants who do or do not meet the World Health Organization's recommendation of an equivalent of 150 minutes of moderate intensity physical activity.

**Methods**. We used a mixed-method design called Group Concept Mapping. In a first phase, we collected statements completing the prompt "In an app that helps me move more, I would like to see/ do/ learn the following…" during four brainstorming sessions with physically inactive individuals ($n = 19$). The resulting 90 statements were then sorted and rated by a new group of participants ($n = 46$). Sorting data was aggregated, and (dis)similarity matrices were created using multidimensional scaling. Hierarchical clustering was applied using Ward's method. Analyses were carried out for the entire group, a subgroup of active participants and a subgroup of inactive participants. Explorative analyses further investigated ratings of the clusters as a function of activity level, gender, age and education.

**Results**. Six clusters of statements were identified, namely 'Ease-of-use and Self-monitoring', 'Technical Aspects and Advertisement', 'Personalised Information and Support', 'Motivational Aspects', 'Goal setting, goal review and rewards', and 'Social Features'. The cluster 'Ease-of-use and Self-monitoring' was rated highest in the overall group and the active subgroup, whereas the cluster 'Technical Aspects and Advertisement' was scored as most relevant in the inactive subgroup. For all groups, the cluster 'Social Features' was scored the lowest. Explorative analysis revealed minor between-group differences.

**Discussion**. The present study identified priorities of users for an mHealth application that promotes physical activity. First, the application should be user-friendly and accessible. Second, the application should provide personalized support and information. Third, users should be able to monitor their behaviour and compare their current activity to their past performance. Fourth, users should be provided autonomy within the app, such as over which and how many notifications they would like to receive, and whether or not they want to engage with social features. These priorities can serve as guiding principles for developing mHealth applications to promote physical activity in the general population.

Corresponding author
Maya Braun, maya.braun@ugent.be

# INTRODUCTION

Promoting physical and mental health is a priority for achieving equity and improving quality of life (*United Nations, 2018*). However, promoting health is a resource-intensive process, and healthcare systems are rarely sufficiently equipped or staffed to fulfill the needs of the population (*World Health Organization, 2018*). Digital health interventions are a promising avenue for the promotion of physical and mental health. They are assumed to be cost-effective and require fewer direct interactions between users and medical staff (*Broekhuizen et al., 2012*). Advantages digital health interventions are that they can be tailored in both their form and their content (*Brug, Oenema & Campbell, 2003*; *Oenema et al., 2008*) and are easily available to a wide part of the public (*De Nooijer et al., 2005*; *Michael & Cheuvront, 1998*). Different reviews have suggested that digital health interventions are promising in promoting physical activity in different target groups (*Rose et al., 2017*; *Stockwell et al., 2019*). Mobile health (mHealth) interventions are a form of digital health interventions making use of smartphones or other mobile technology, providing opportunities to expand the reach of health interventions further (*Kumar et al., 2013*), while remaining effective in different target groups (*Aslam et al., 2020*; *Mönninghoff et al., 2021*).

Iteratively involving end-users throughout the development of digital interventions is widely considered critical to achieve effective engagement (*Pagliari, 2007*; *Yardley et al., 2015*). One of the notable frameworks in digital intervention development is the person-based approach (*Yardley et al., 2015*). This approach was developed specifically for digital interventions and consists of two elements: involving qualitative research with people from the target user population at each stage of intervention development, and providing guiding principles that can inspire and inform development. This approach is complementary to theory-based and evidence-based approaches, which also inform development (*Yardley et al., 2015*). Likewise, van Gemert-Pijnen and colleagues have created a holistic framework for participatory development, including end-users (*Van Gemert-Pijnen et al., 2011*). Van Velsen and colleagues also called for an extensive involvement of end-users in the development of interventions *via* qualitative studies, such as interviews, focus groups and observations, aiming to identify and understand the needs of the target group (*Van Velsen, Wentzel & Van Gemert-Pijnen, 2013*). More generally, involving end-users and other stakeholders within development has been identified as key to achieve a technology that is truly user-informed (*Pagliari, 2007*).

Different methods can be used to involve end-users. Studies in the domain of physical activity (PA) promotion have often used end-users as part of the process evaluation of interventions, which has informed the development and adaptation of later (versions of) interventions. Semi-structured interviews are frequently used after a period of using the intervention in order to gain further insights into the usability and feasibility (*Baretta,*

*Perski & Steca, 2019*; *Degroote et al., 2020*; *Pollet et al., 2020*). While this allows for user-input that can inform the development and adaptation of later (versions of) interventions, it does not involve end-users in the development process itself. However, there are other examples of studies that have made use of participatory design approaches, including think-aloud interviews (*Pollet et al., 2020*; *Poppe et al., 2017*; *Rowsell et al., 2015*), prototype activities (*Van Hierden, Dietrich & Rundle-Thiele, 2021*), individual and group (feedback) interviews (*Ehn et al., 2021*; *Van Hierden, Dietrich & Rundle-Thiele, 2021*), focus groups (*Heffernan et al., 2016*), informal guerilla testing (*Heffernan et al., 2016*) and advisory Facebook groups (*Heffernan et al., 2016*).

Finding ways to meaningfully engage patients and other stakeholders in the design process is one of the challenges in intervention development (*Voorheis et al., 2022*). Most studies aiming to include stakeholder input use either a qualitative (*Bergevi et al., 2022*; *Poppe et al., 2018*) or quantitative (*Guertler et al., 2015*; *Schroé et al., 2022*) approach. Group Concept Mapping (GCM) attempts to unite these two approaches by including user participation in two phases (*Trochim & Kane, 2005*), one phase about generating statements (qualitative) and a second phase about sorting and rating those statements (quantitative).

GCM results in labelled clusters of statements, rated by participants based on urgency, perceived importance or other relevant values. It has been developed to integrate input from multiple sources with differing content expertise or interest. It allows for direct comparison of the ratings provided by different groups, such as different kinds of stakeholders or subgroups within the end-users. The resulting maps of statements and clusters provide a structure that can easily be used to guide intervention development (*Trochim & Kane, 2005*).

GCM has been used in the domain of PA in various groups, including different age groups (*Baskin et al., 2015*; *Hanson et al., 2013*), and different patient groups (*Fitzpatrick & Zizzi, 2014*; *Strassheim et al., 2021*; *Vlot-van Anrooij et al., 2020*). Notably, research often focused on identifying benefits, action steps, barriers or risk factors rather than the involvement in the intervention development process. Hence, the present study used a GCM approach in order to gain insights into the needs and wishes of potential users of digital interventions promoting PA. We wanted to answer the following two questions:

(1) What do users need or wish in a smartphone application that would help them be more physically active?

(2) Are there differences in wishes and needs as a function of user characteristics? Specifically, do users who regularly meet the World Health Organization's PA guidelines of 150 min of moderate activity a week report different needs and wishes than users who do not meet the norms.

## MATERIAL AND METHODS

We followed the GCM approach introduced by Trochim and Kane (*Trochim, 1989*; *Trochim & Kane, 2005*). We structured our approach into two phases: the idea generation phase, and the structuring phase. Below, phases are described as two separate studies, as the samples of participants in the two studies were independent.

## Phase 1: generation of statements
### Participants

Participants were recruited using flyers and social media advertisement in November and December of 2021 in Ghent, Belgium. Flyers were laid out at different university sites, healthcare facilities, stores and the public library. Social media advertisement was placed in regional Facebook groups of different areas of Ghent, on the researchers private social media and on the research group's social media. Paid ads were used for the latter. Each participant that completed the session received a €20 incentive.

In total, 133 people completed the screening questionnaire, of which 59 met the inclusion criteria concerning age (between 18 and 65), health (being able to walk at least 100m without aids) and current PA level (below the WHO recommendations of 150 min of moderate activity a week). Of these, 21 took part in four online brainstorming sessions, and 19 remained until the end of the online session. Two participants dropped out due to technical or practical difficulties. Two brainstorming sessions had four participants, one had six (five after one dropping out), and one had seven (six after one participant dropped out).

Within the group of participants was one man (5.26%). Eleven participants (57.89%) were 50 years or older, eight (42.11%) were between 26 and 49 years old. Nine participants (47.37%) had university level education, five (26.32%) had finished non-university higher education, four (21.05%) had finished secondary school (of which three technical and one general), and one (5.26%) had finished lower education.

Concerning their PA in the previous week, participants reported a median of 313.82 MET minutes (SD = 191.43), ranging from 0 to 577.5 MET minutes. This is equivalent to an average of 78.4 min of moderate activity, 39.22 min of intense activity or 95 min of walking.

All participants provided written informed consent to participate. The study conformed with the general ethical protocol for scientific research at the faculty of psychology and educational sciences of Ghent University.

### Material

We used MIRO (https://miro.com/) to create digital post-it notes notes in order to facilitate discussion. Screenshots of both the introductory exercises used, and the frame used for discussion, can be found in File S1.

Participants filled in a screening questionnaire including questions about their age, highest level of education, physical health (specifically whether they were able to walk without walking aids for at least 100 m) and current PA habits. For current PA behaviour, we used the International Physical Activity Questionnaire (IPAQ, *Vandelanotte et al., 2005*). This questionnaire assesses the amount of vigorous, moderate activity and walking activity that a participant has performed in the week prior to assessment.

### Procedure

The generation of ideas was done in brainstorming sessions in group. While the brainstorming sessions were originally planned to take place in person using post-its

for brainstorming, the research team decided to switch to online meetings due to the COVID-19 pandemic.

The general structure of the brainstorming sessions is summarized in Fig. 1. All sessions were led by MB and supported by SC and MVDV. In short, the session consisted of an introduction where the researchers introduced themselves, and the participants were introduced to the topics of PA and mHealth. Afterwards, there was an introductory exercise to make sure all participants were sufficiently familiar with the program for brainstorming. This also served as the basis for a short introduction round with the participants. An individual brainstorming followed where participants completed the prompt: "In an app that helps me move more, I would like to see/learn/do the following: …" by writing their answers on separate digital post-its. After a short break, a group discussion followed based on the post-its written. In the group discussion, ideas could be further clarified and other participants had the opportunity to add their own related statements. New ideas could be introduced during the group session, and were then written onto post-its either by one of the researchers or the participants themselves. Post-its were loosely arranged into labelled groups during discussion to make it easier for participants and researchers to keep an overview during the discussion. The session lasted approximately 2 h each. Next to audio recording the statements of participants during the brainstorming sessions using digital post-its, sessions were also audio recorded and could be referred back to throughout data processing.

## Data analysis

The statements created during the brainstorming sessions were processed by a researcher who attended the brainstorming sessions in the following ways: first, statements that were either exact duplicates or communicated the same idea were grouped, and one statement was chosen to represent this group. For example, the statements "not expensive", "a free app" and "no subscription fees" were combined into the statement "no costs". Second, if necessary, that statement was rephrased to match the original prompt ("In an app that helps me move more, I would like to see / learn / do the following …") if it did not already. For example, the statement "helps me move without much additional material" was rephrased to "support in moving more without requiring additional material".

Processing of the statements took place after all brainstorming sessions were finished. The resulting list of statements was checked and approved by all researchers who attended the focus groups (MB, SC, MVDV), who compared the original statements from the brainstorming session with the resulting statements in order to stay as close to the original ideas as possible. The list was revised based on their feedback. Moreover, clarity of the resulting statements was checked by researchers who had not attended the brainstorming session, and did not have access to the original statements, by providing them with a written list of statements (ADP, GC, FDB).

## Phase 2: sorting and rating of statements
### Participants
Dutch-speaking participants were recruited using Prolific (https://www.prolific.co/). The sorting and rating task was completed and considered valid by 48 and 51 participants,

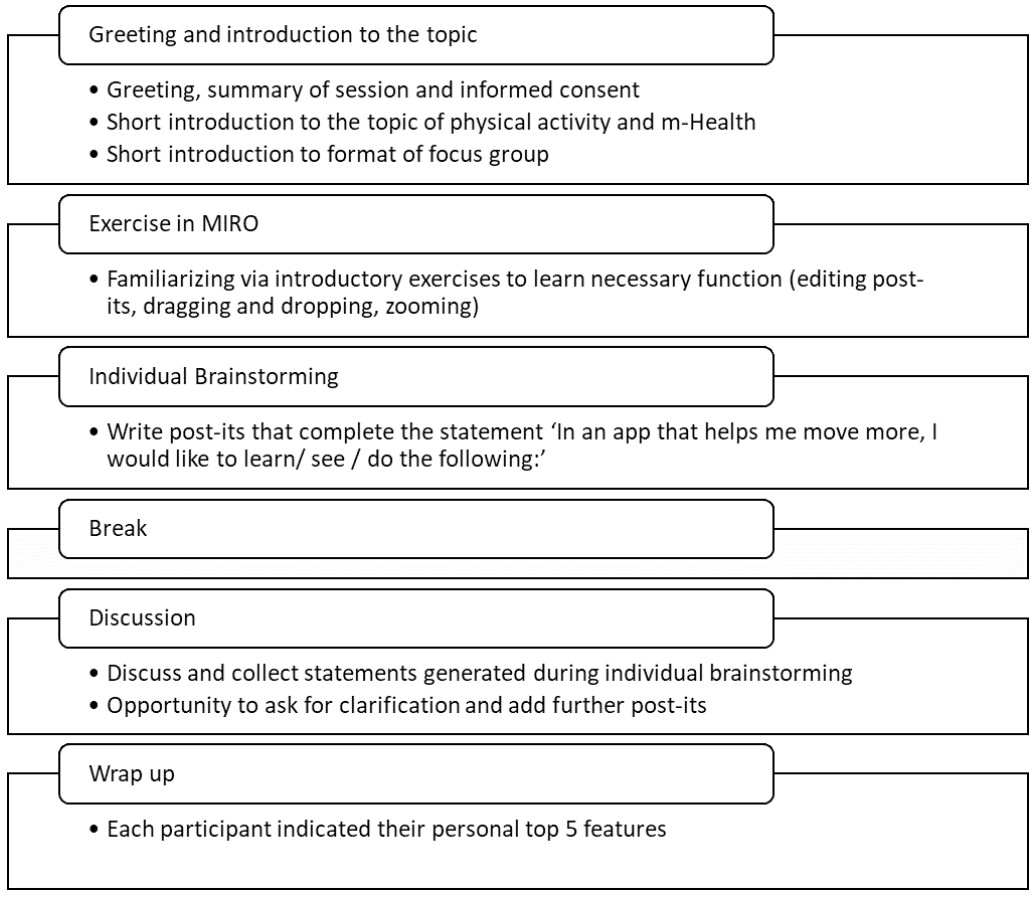

**Figure 1** Overview of the brainstorming sessions.

respectively. The study was estimated to take 45 min, and participants received $6 as an incentive.

A total of 52 different participants provided valid data for at least one of either sorting or rating task, but only 51 (23 women) provided sociodemographic information. Further information on why some data was excluded can be found in the data analysis section. Age ranged from 19 to 61, with a mean of 30.61 and a median of 28 (Q1 23.5 Q3 35.5). Most participants reported to have completed university level education ($n = 25$). Higher non-university education, higher secondary education and higher vocational education were reported by six, 11 and nine participants respectively. About half of the participants reported that they have been active for a long time ($n = 24$). Ten participants reported to have been active for a short time, whereas 17 participants reported not to be active. Out of the latter, four were contemplating about starting be active, seven had the intention to start and six tried to start but did not succeed yet.

Participants gave written informed consent to participate in the study. The study was approved by the ethical committee of the faculty of psychology and educational sciences of Ghent University in January 2021 (reference 2021-198).

### Material

We made use of the groupwisdom™ software (The Concept System® *GroupWisdom, 2022*). Groupwisdom™ is a tool specifically designed for GCM studies. While it allows for both the idea generation phase and the sorting and rating phase to be completed within the tool, we opted to only use it for the sorting and rating phase.

### Procedure

Phase 2 took place online with no direct interaction between researchers and participants, though participants were able to pose questions *via* e-mail. Participants filled in a short questionnaire providing information on their age, gender, education level and current PA level. PA level was assessed with one question asking whether participants were following the World Health Organization's guidelines of 150 min of moderate activity per week. Participants could indicate whether (1) they had no intention of following the guidelines, (2) they were contemplating following them, (3) they had the intention to start following them, (4) they had the intention but were not able to do it, (5) they had been doing it for a short while, or (6) that they had been doing it for a long while.

They then sorted the existing statements into labelled piles based on the content of the statements. The specific way they sorted the statements was decided upon by the participants themselves and was not predetermined by the research team. Participants also determined the amount of piles they wanted to create, though they were asked not to create more than 20 piles. We further asked participants to sort by topic, not by values such as 'most important' or 'essential' or 'helpful'. After sorting, participants were asked to rate each individual statement ("How important do you find it that you can learn, see of do this in an app that helps you move more?") on a scale from 1 ("not important at all") to 5 ("very important"). In both tasks, statements were presented in random order.

## Data analysis

The analysis and visual representation of statements was mainly done using groupwisdom™. First, the validity of all data was checked by the primary researcher (MB): Sorting data was considered invalid if there were only two or less categories ($n = 2$), or more than 20 categories ($n = 1$), the categories were based on importance, relevance or priority (*e.g.*, piles called "the app needs this", "unnecessary features", $n = 3$). Unnamed piles were not accepted into data analysis if the primary researcher (MB) judged that the sorting was not based on the content of the statements, but rather random ($n = 1$). In total, 59 started the sorting task, 55 completed it (*i.e.*, sorted at least 75% of statements into categories) and 48 participants provided valid data. A total of 52 participants started the rating task, and 51 rated at least 75% of the statements. We also planned to exclude data when observing a pattern in the answers (*e.g.*, only answering 3, following a pattern of $1 - 2 - 3 - 4 - 5$ while answering), but this was not the case.

The sorting data was aggregated in groupswisdom, and a similarity and dissimilarity matrix were created. Multidimensional scaling was used in order to determine which statements are more or less similar to each other based on how often they were sorted together. The results of this step are depicted in a point map. Following this, hierarchical

cluster analysis was applied using Ward's method. This way, statements that are most similar were grouped together in clusters. As groupwisdom does not offer further support in choosing a cluster solution, we instead used the cluster package in R (*Maechler et al., 2022*). This package allows the use of multidimensional scaling and hierarchical clustering, just as it is done in groupwisdom. We then plotted the eigenvalues of the different cluster solutions, using the *elbow* in the scree plot as a minimum amount of clusters. All cluster solutions containing at least that amount of clusters and at most 20 clusters were evaluated by the primary researcher (MB) by reviewing the content of the statements belonging to each cluster. The goal was to find a cluster solution where (a) statements within one cluster match an underlying theme as well as possible and (b) there is little overlap between different clusters. Solutions with fewer clusters were further prioritized over solutions with many clusters. The three most suitable cluster solutions were presented within the research group, and a final cluster solution was chosen. The clusters and their statements were presented to each member of the research team who then independently chose appropriate labels for each cluster. The primary researcher (MB) finalized the list of labels based on the labels provided by the research team and the labels suggested by participants for similar clusters throughout the sorting procedure. Rating data was aggregated on both the statement and the cluster level.

Analyses were first performed for the entire sample, and subsequently repeated for two subgroups: participants who indicated that they have been physically active according to the WHO guidelines for a longer period, corresponding to answer "Yes, for a long while" ($n = 22$, further called active participants) and participants who indicated that they have not been regularly following the WHO guidelines for PA, corresponding to answers "No, and I am not planning to start", "No, but I am considering it", "No, but I have the intention to start", "No, I have the intention to start, but it does not work", or they have only been doing so for a short time, corresponding to answer "Yes, for a short while" ($n = 24$, further called inactive participants). Further explorative analyses were repeated for the following subgroups: male ($n = 22$) and female ($n = 23$) participants, participants who have completed higher education ($n = 24$) and those who have not ($n = 24$), three approximately equally sized age groups, namely up to age 24 ($n = 16$), age 25–31 ($n = 15$) and older than 31 ($n = 15$). To explore those subgroups further, independent two-sample t-tests were executed per cluster to test whether the ratings of the clusters differed between subgroups. Two-sided uncorrected *p*-values were used to determine statistical significance at an alpha level of 0.05 (*Rothman, 1990*). In addition, *p*-values were compared to the alpha value divided by the number of tests in order to account for multiple testing using the Bonferroni method.

## RESULTS

### Phase 1: generation of statements

In the brainstorming sessions, a total of 373 post-its were created, with an average of 93 (SD = 27) post-its per group, ranging from 60 to 125 post-its per group. After processing, the final list contained 90 statements.
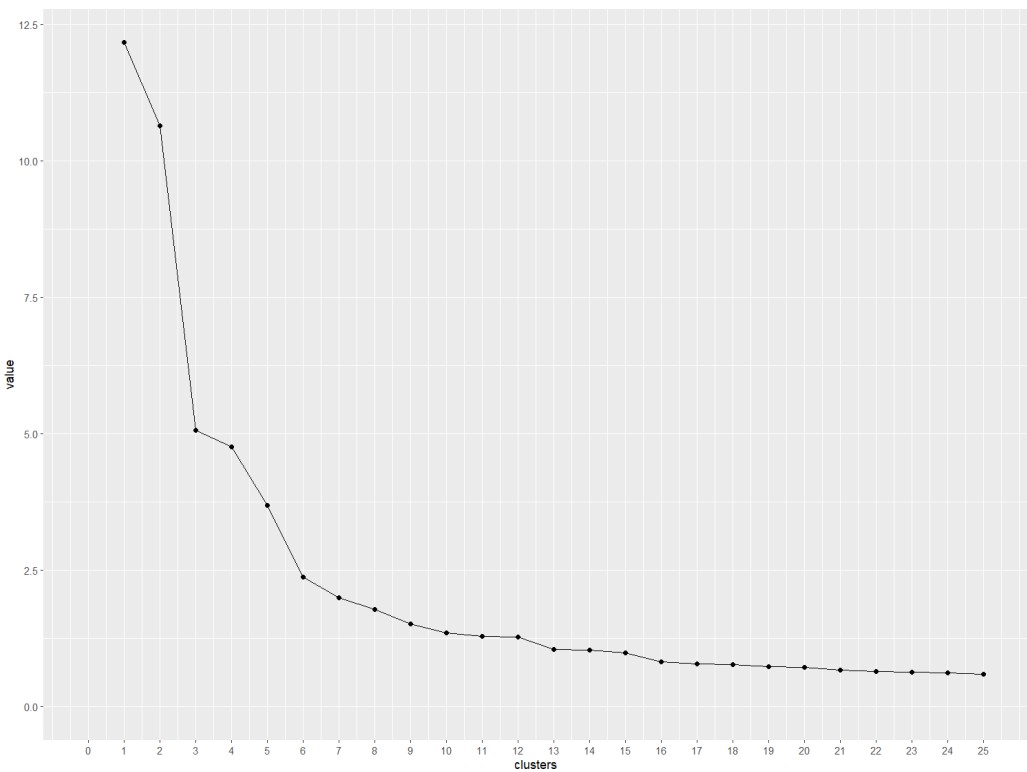

**Figure 2** **Scree plot presenting the eigenvalues of cluster solutions with 1–25 clusters.**

## Phase 2: sorting and rating
### Cluster structure

The scree plot in Fig. 2 indicates a cluster structure with at least six clusters, with limited added value after 10 clusters. In order to reach a final cluster solution, the primary researcher (MB) evaluated all cluster solutions between six and ten clusters by reviewing the statements belonging to each cluster. The most suitable cluster solutions had six and eight clusters respectively. They were chosen based on how strongly statements within one cluster formed a group that is separate from the other clusters, and were then discussed in the research team. A solution with six clusters was chosen, as depicted in Fig. 3.

The first identified cluster was labelled '*Ease-of-use and self-monitoring*', and contained 12 statements regarding the accessibility and ease of use of the application (*e.g.*, "Texts and figures should be easily understood") as well as self-monitoring aspects (*e.g.*, "I want to automatically see how much I've moved in a day (without having to enter it)"). The cluster 'Personalized *information and support*' consists of 30 statements referring to ways in which an mHealth application can provide support based on the individual and their context (*e.g.*, "Suggestions for activities that take pain or ailments into account", "Suggestions for activities that take into account where I am"), providing health-related information (*e.g.*, "Information on how my movement affects my health"), and more general support (*e.g.*, "Support in dealing with obstacles and barriers", "Support to replace or interrupt seated activities, such as at work"). The cluster '*Motivational Aspects*' consists of 11 statements

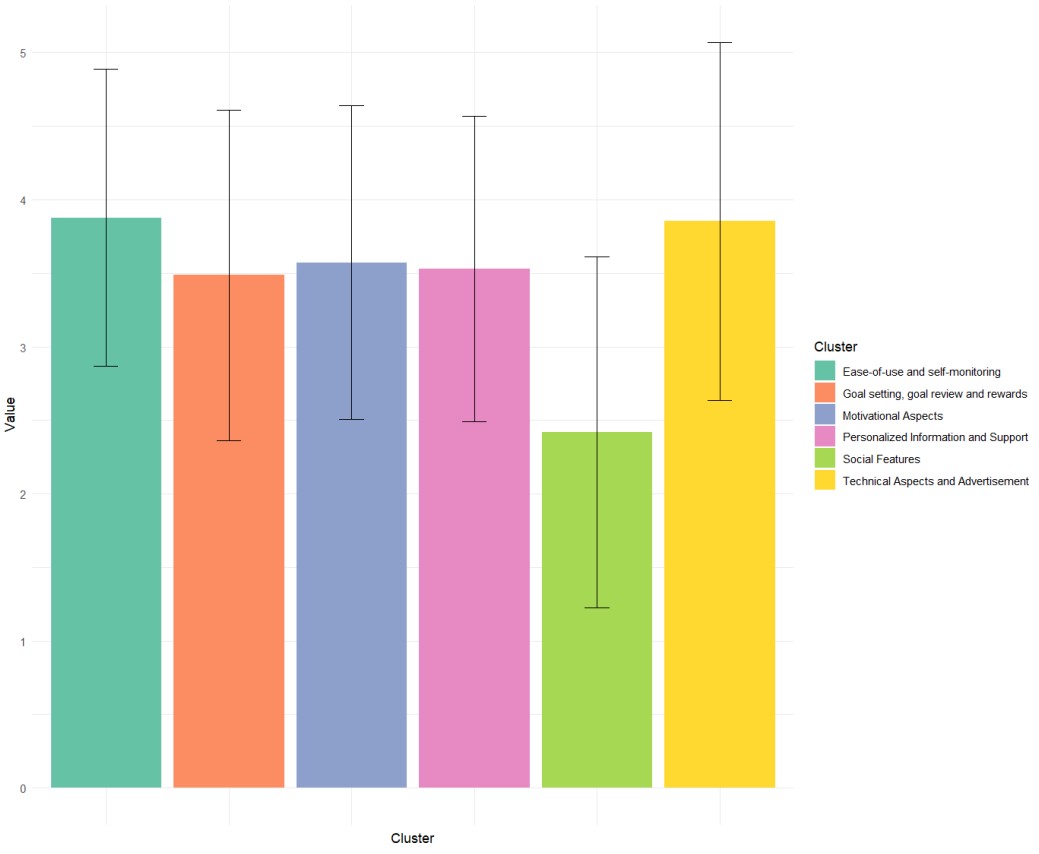

**Figure 3** **Bar chart comparing the average importance ratings of each cluster.** Error bars represent standard deviation of ratings within each cluster.

referring to positive or motivating messages (*e.g.*, "Motivational messages", "Getting compliments and affirmations after moving"). The cluster *'Goal Setting, Goal Review and Rewards'* consists of 12 statements referring to setting and reviewing goals (*e.g.*, "Set up short-term goals for myself"), and receiving rewards if they are reached (E.g. "Getting rewards within the app such as badges, symbols, streaks or points"). The cluster *'Technical Aspects and Advertisement'* consists of 12 statements referring to the business model of the application (*e.g.*, "No costs" or "Only ads that don't bother me while using the app (*e.g.*, No waiting times for ads)"), as well as technical aspects (*e.g.*, "Links to a digital agenda"). The final cluster, *'Social Features'*, contained 13 statements regarding interactions with others within the app (*e.g.*, "I want to be able to chat with others", "I want to see other people's activity in the app").

There was some overlap between clusters 'Goal Setting, Goal Review and Rewards' and 'Motivational Aspects' concerning rewards. Whereas the statements of the 'Motivational Aspects' cluster often remained relatively abstract (*e.g.*, "Game elements that motivate me to move more"), the statements regarding rewards in the cluster 'Goal Setting, Goal Review and Rewards' directly linked rewards to reaching specific goals (*e.g.*, "Setting up my own

rewards that I give to myself when I reach a certain goal'', ''Rewards when I move more or reach my goal'').

A list of all clusters and their corresponding statements can be found in File S3.

### Rating of importance

The ratings of importance for each cluster by the complete groups and all subgroups can be found in Table 1. A map of the final cluster structure can be found in Fig. 4.

*Statements*

On average, participants gave a score of 3.42 (SD = 0.68) with the lowest average rating of 1.65 for 'sharing photos with others' and ''chatting with others'' and a highest average rating of 4.76 for ''being able to use the app easily''. Active participants gave an average score of 3.45 (SD = 0.67) with a lowest average rating of 1.73 for ''sharing photos with others'' and a highest average rating of 4.82 for ''being able to use the app easily'', and inactive patients an average score of 3.38 (SD = 0.72) with a lowest average rating of 1.46 for ''being able to chat with others'' and a highest average rating of 4.71 for ''being able to use the app easily''.

*Clusters*

The average rating of importance for clusters was 3.42 (SD = 0.55) with the lowest rating of 2.36 for 'Social Aspects' and the highest rating of 3.86 for 'Ease-of-use and Self-monitoring' for the total group. For the 'active' subgroup, the average was 3.37 (SD = 0.62) with the lowest rating of 2.16 for 'Social Aspects' and the highest rating of 3.86 for 'Technical Aspects and Advertisement'. For the inactive subgroup an average of 3.45 (SD = 0.61) with the lowest rating of 2.23 for 'Social Aspects' and the highest rating of 3.86 for 'Technical Aspects and Advertisement'. Importance ratings of all clusters for each subgroup can be found in Table 1.

*Explorative analysis: comparison between subgroups*

All subgroups rated the clusters 'Ease-of-use and Self-monitoring' and 'Technical Aspects and Advertisement' highest with values ranging from 3.74 to 3.96. Which one of the two was rated higher. varied, with the group of inactive participants, the youngest age group (aged up to 25 years), the group that finished higher education and the group of men rating 'Technical Aspects and Advertisement' highest. All groups rated 'Social Features' lowest, with values ranging from 2.16 to 2.48. The ratings of the remaining clusters ranged from 3.27 to 3.66, with different orders in different subgroups. For a full list of all statements and clusters and their respective ratings by each groups, refer to the File S4.

A comparison of cluster ratings between different subgroups can be found in Fig. 4, with part (a) comparing active and inactive participants, part (b) comparing the different age groups, part (c) comparing different educational levels and part (d) comparing different genders. Pairwise comparisons were conducted to compare the ratings of individual clusters between groups at an alpha level of 0.05.

Active people rated statements from the cluster 'Social Features' significantly higher in importance than inactive people ($t$ (636.19) = 3.55, $p$ = 0.001). Women rated 'Ease-of-use

**Table 1  Pairwise comparison of cluster ratings between subgroups.**

| Cluster | Mean (SD) of subgroup 1 | Mean (SD) of subgroup 2 | *p*-value |
|---|---|---|---|
| **Pairwise comparison: Active ($n = 22$) vs inactive ($n = 24$) subgroup** | | | |
| Ease-of-use and self-monitoring | 3.9(1.07) | 3.86(0.96) | 0.67 |
| Goal setting, goal review and rewards | 3.5(1.15) | 3.48(1.1) | 0.84 |
| Technical Aspects and Advertisement | 3.81(1.24) | 3.89(1.2) | 0.4 |
| Social Features | 2.59(1.23) | 2.26(1.14) | **<0.01** |
| Personalized Information and Support | 3.51(1.07) | 3.55(1.01) | 0.39 |
| Ease-of-use and self-monitoring | 3.9(1.07) | 3.86(0.96) | 0.67 |
| **Pairwise comparison: male ($n = 22$) vs female ($n = 23$) subgroup** | | | |
| Ease-of-use and self-monitoring | 3.8(1.1) | 3.96(0.9) | 0.05 |
| Goal setting, goal review and rewards | 3.44(1.23) | 3.54(0.99) | 0.28 |
| Technical Aspects and Advertisement | 3.83(1.24) | 3.86(1.21) | 0.78 |
| Social Features | 2.38(1.21) | 2.44(1.18) | 0.57 |
| Personalized Information and Support | 3.51(1.06) | 3.55(1.02) | 0.39 |
| Motivational Aspects | 3.48(1.14) | 3.66(0.97) | 0.05 |
| **Pairwise comparison: subgroup with ($n = 24$) vs without ($n = 24$) higher education** | | | |
| Ease-of-use and self-monitoring | 3.87(1.02) | 3.89(0.99) | 0.78 |
| Goal setting, goal review and rewards | 3.53(1.1) | 3.42(1.15) | 0.22 |
| Technical Aspects and Advertisement | 3.91(1.18) | 3.76(1.28) | 0.13 |
| Social Features | 2.38(1.19) | 2.47(1.2) | 0.36 |
| Personalized Information and Support | 3.5(1.04) | 3.58(1.04) | 0.12 |
| Motivational Aspects | 3.6(1.05) | 3.54(1.1) | 0.55 |
| **Pairwise comparison: Agegroup 1 ($n = 16$) and Agegroup 2 ($n = 15$)** | | | |
| Ease-of-use and self-monitoring | 3.88(1.06) | 3.84(1.05) | 0.77 |
| Goal setting, goal review and rewards | 3.61(1.11) | 3.36(1.21) | **0.03** |
| Technical Aspects and Advertisement | 3.9(1.25) | 3.8(1.24) | 0.43 |
| Social Features | 2.32(1.17) | 2.48(1.22) | 0.18 |
| Personalized Information and Support | 3.63(1.05) | 3.41(1.04) | **<0.01** |
| Motivational Aspects | 3.6(1.07) | 3.42(1.15) | 0.13 |
| **Pairwise comparison: Agegroup 1 ($n = 16$) and Agegroup 3 ($n = 15$)** | | | |
| Ease-of-use and self-monitoring | 3.88(1.06) | 3.91(0.93) | 0.74 |
| Goal setting, goal review and rewards | 3.61(1.11) | 3.47(1.06) | 0.17 |
| Technical Aspects and Advertisement | 3.9(1.25) | 3.85(1.18) | 0.69 |
| Social Features | 2.32(1.17) | 2.46(1.2) | 0.23 |
| Personalized Information and Support | 3.63(1.05) | 3.53(1.01) | 0.14 |
| Motivational Aspects | 3.6(1.07) | 3.68(0.98) | 0.41 |
| **Pairwise comparison: Agegroup 2 ($n = 15$) and Agegroup 3 ($n = 15$)** | | | |
| Ease-of-use and self-monitoring | 3.84(1.05) | 3.91(0.93) | 0.53 |
| Goal setting, goal review and rewards | 3.36(1.21) | 3.47(1.06) | 0.36 |
| Technical Aspects and Advertisement | 3.8(1.24) | 3.85(1.18) | 0.67 |
| Social Features | 2.48(1.22) | 2.46(1.2) | 0.83 |
| Personalized Information and Support | 3.41(1.04) | 3.53(1.01) | 0.06 |
| Motivational Aspects | 3.42(1.15) | 3.68(0.98) | **0.02** |

**Notes.**
Bold values indicate statistical significance at an alpha-level of 0.05.

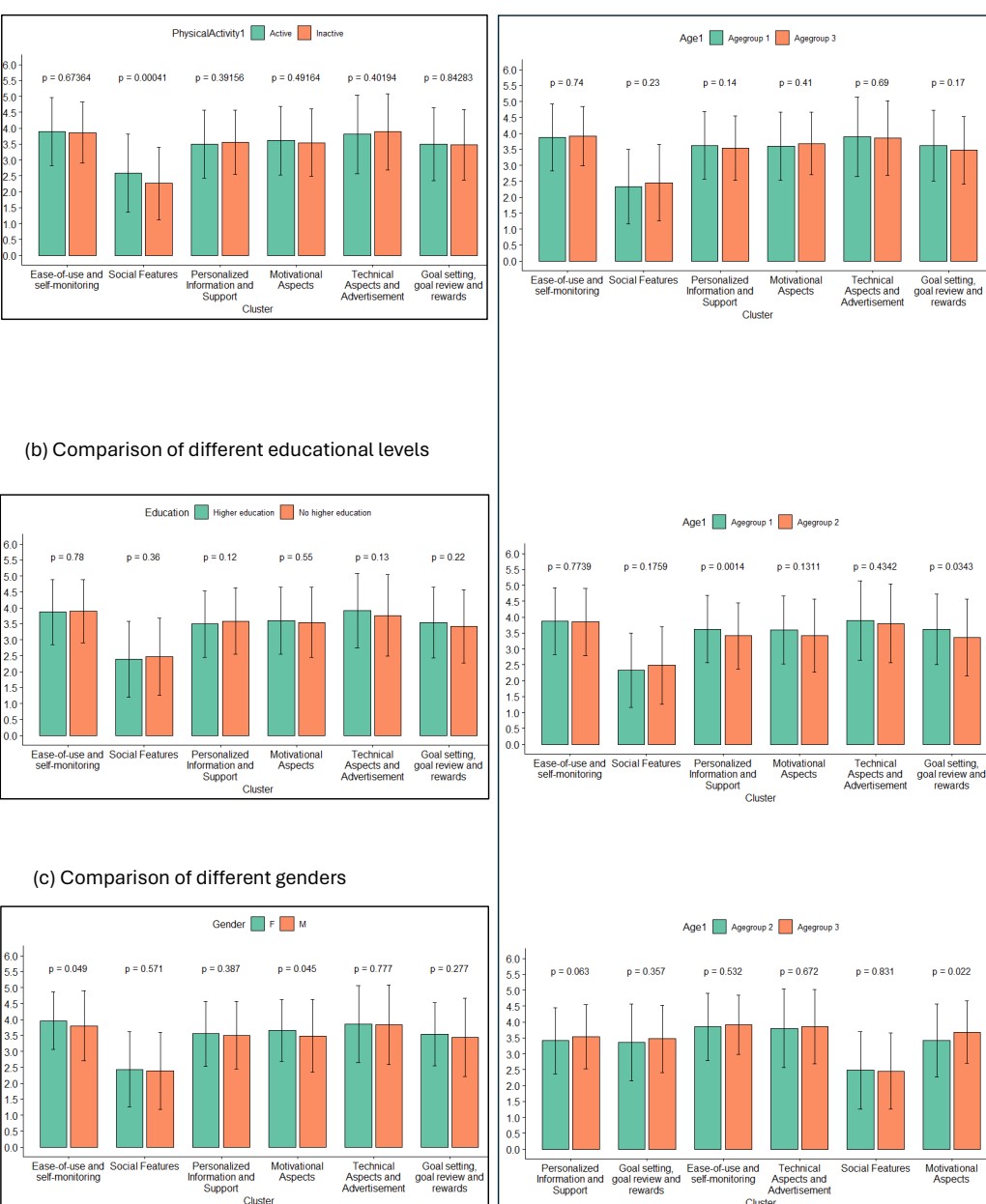

**Figure 4  Bar charts comparing the cluster ratings between two different groups.** The rating of each cluster is compared between two subgroups, with (A) comparing active and inactive participants, (B) comparing three different age groups, (C) comparing participants with higher education to those without and (D) comparing male and female participants. Individual statements were rated on a scale from 1 to 5. Error bars represent standard deviation within each respective group for a given cluster.

and Self-monitoring' ($t$ (597.20) $=-1.97$, $p = 0.049$), and 'Motivational Aspects' ($t$ (546.95) $=-2.01$, $p = 0.045$) higher than men.

There are significant differences in ratings of importance for both 'Goal Setting, Goal Review and Rewards' ($t$ (366.55) = 2.12, $p$ = 0.034) and 'Personalized Information and Support' ($t$ (958.07) =3.20, $p$ <0.001) between participants under 25 and those between 25 and 31, and for 'Motivational Aspects' between participants between 25 and 31 and over 31 ($t$ (322.69) = −2.31, $p$ = 0.022). It should be noted that a majority of the significant effects disappear if we perform Bonferroni alpha correction by dividing the alpha level by 6, the amount of independent subgroup comparisons, resulting in an alpha level of 0.0083. All pairwise comparisons are depicted in Table 1.

## DISCUSSION

We investigated the needs and wishes of potential users of an mHealth application that aims to promote PA in the general population using a mixed-method approach (GCM). In a first study, 22 participants provided ideas on what they would like to see, learn, or do in such an application. This resulted in a final set of 90 statements. In a second study 52 participants sorted each statement and rated its importance. The chosen cluster structure contained six clusters: 'Ease-of-use and Self-monitoring', 'Technical Aspects and Advertisement', 'Motivational Aspects', 'Personalized Information and Support', 'Goal Setting, Goal Review and Rewards', and 'Social Features' (sorted by rating).

The features identified can be mapped on the persuasive system design framework (Oinas-Kukkonen & Harjumaa, 2009), which provides design methods for developing persuasive software solutions. This framework classifies features that support behavioral change into four distinct categories: *primary task support, dialogue support, credibility support and social support*. Most of the highly rated features (rated "rather important" or "very important") within this study fit under *primary task support*. Primary task support directly supports the change of the target behaviour, such as through tailoring, personalization, self-monitoring, or improved ease-of-use.

Concerning tailoring and personalization, users in the study by Degroote and colleagues (Degroote et al., 2020) suggested receiving personalized suggestions for action and coping plans in order to improve the ease-of-use of the application. More broadly, applications and their content being tailored to the user groups and personalized to individual users is frequently desired (Bergevi et al., 2022; Yan et al., 2015), and has been found to be more effective than generic interventions in meta-analyses (Noar, Anderson & Harris, 2007) and systematic reviews (Broekhuizen et al., 2012), though the strength of the evidence is not yet clear (Conway et al., 2017). Tailoring is generally considered an intervention characteristic that is associated with higher user engagement (Vandelanotte et al., 2016), and has been recommended in intervention design approaches such as the person-based approach (Yardley et al., 2015). Other statements that were highly rated by participants concerned the agency about the kind and frequency of notifications within the application ("Set what type of notifications I want to receive or not", "Set when I want to receive notifications or not."). This may also be considered a kind of personalization. In the person-based approach, Yardley and colleagues stress the need for autonomy of the user in intervention development. They specifically suggest leaving it up to the user at what times they receive what kind of notifications (Yardley et al., 2015).

Some of the highest rated statements concerned self-monitoring one's PA ("See how much I've moved at the end of the week", "See how much I've moved at the end of the day", "Being able to compare my exercise performance with older data about myself"). This matches previous quantitative and qualitative findings. Self-monitoring of behaviour has been found to be an effective behavior change technique (BCT) to achieve behavior change in regards to PA (*Greaves et al., 2011*; *Michie et al., 2009*), alongside other self-regulatory techniques. It is also among the BCTs that are most commonly used in publicly available apps that focus on PA (*Bondaronek et al., 2018*). Monitoring and feedback have also been reported as an important feature of an mHealth app focusing on PA by other populations, such as college students (*Yan et al., 2015*), HIV patients (*Montoya et al., 2015*), breast cancer survivors (*Phillips et al., 2019*) or multiple sclerosis patients (*Giunti et al., 2018*), especially when combined with wearables to automatize monitoring.

When it comes to the aspect of ease-of-use, our findings are corroborated by a substantial body of qualitative research.. For example, users in a study by *Degroote et al. (2020)* suggested to reduce text input in order to make an mHealth application less time-intensive and user friendly. Further, both participants from a general population (*Bergevi et al., 2022*) and patients with breast cancer (*Phillips et al., 2019*) or depression (*Avila-Garcia et al., 2019*) have reported that an application being 'easy to use' by being easy to read, engaging and visually appealing is a priority. In a review of the state of the art, *Vandelanotte et al. (2016)* stressed the need for applications to take accessibility into account, especially when it comes to underserved populations, such as those with low-socioeconomic status or low social capital. Based on the high amount of highly-rated statements and clusters relating it, primary task support seems to be the highest priority for persuasive features in an mHealth intervention promoting PA in the general public.

*Dialogue support* deals with the feedback that the system offers in guiding the user to reach the intended behaviour. Multiple features relating to it were mentioned in the statements. Specifically, praise (*e.g.*, "Getting compliments and affirmations after moving"), rewards (*e.g.*, "Getting rewards within the app such as badges, symbols, streaks or points.") and reminders (*e.g.*, "Notifications that remind me of my planned activities just before starting the activity") were mentioned, especially in the clusters 'Motivational Aspects' and 'Goal setting, goal review and rewards'. There was a considerable amount of statements created referring to features relevant for dialogue support, with none receiving low ratings. For the development of an activity promotion mHealth intervention in a general population, we would thus consider dialogue support the second highest priority, especially focusing on rewards, praise and reminders.

*Credibility support* improves how trustworthy the system appears to the end-user. This dimension was only mentioned twice in the statements "Connection to an institution I can trust, such as a hospital or university", and "Periodic follow-up of my progress on the app by a counsellor or expert.". Neither of those statements were of particular priority to participants of the current study, both receiving below average ratings. Hence, users did not consider credibility support a high priority for mHealth interventions promoting PA. There is a number of reasons that could explain the gap between the general advice to integrate credibility support and the results we have found. First, the sample in this

study consisted of generally healthy individuals wanting to increase their physical activity. While credibility support in general is not specific to clinical samples, it might be perceived as less important for those who are not currently experiencing any symptoms or pain. This might specifically be the case for mHealth interventions targeting PA, as participants might compare those to existing commercial apps, such as Fitbit, which are not linked to existing institutions but work well. Second, the current study was carried out by researchers affiliated with a university. As such, participants might have considered that it was a given that the intervention would be trustworthy, and did not think any credibility support *beyond* affiliation with a university was necessary Similarly, it could be that credibility support is implied in a number of statements that are rated higher. For example, when participants rate 'personalized health information', the underlying assumption could be that the information is trustworthy.

Despite the low ratings credibility support received in this study, the principles of 'trustworthiness' and 'verifiability' should still be followed when developing interventions because some elements of credibility support also concern the scientific base of the intervention, as well as its transparency.

*Social support* concerns any features where users interact with other users. Within the statements in this study, it was covered by the cluster 'Social Features', including social learning, social comparison, social facilitation, cooperation, competition and recognition. Remarkably, social features were rated the lowest for an mHealth applications in our study. This is surprising, as a large number of statements (13 out of 90) about social features were generated in phase 1. Previously, patient groups have stressed the need for social features to be optional (*Giunti et al., 2018*), a view that was also endorsed in our study. Existing research found mixed results concerning the usage of social features, with no significant effect of the usage on intervention outcomes (*Tong & Laranjo, 2018*). However, other researchers have stressed social features, including social comparison and receiving social support, as key aspects for promoting PA, with almost half the participants rating it as useful (*Ayubi et al., 2014*).

There is a number of possible reasons why statements related to social support were rated as relatively unimportant by our target group. First, many of the suggested features, such as "Share photos with others", "Make arrangements with others to exercise together" or "Connect with my friends inside the app" could feel unnecessary to many as it is redundant with available technology , such as messaging apps.

Based on this study, implementing social support components does not seem to be a priority for mHealth interventions targeting physical activity in a healthy adult population. Further research would be required in order to determine what kinds of social support are deemed helpful by users. Open questions concern amongst others whether participants want to interact with existing connections (*e.g.*, friends) or want to meet new people and whether participants want to interact within the intervention or want the intervention to facilitate offline contact. Redundancy with established apps also needs to be considered. Most importantly, users want to be able to have agency about whether or not they use social support features within any program.

### Implications for future research

The current study explored the needs and wishes of a general adult population. Future research could expand upon our research not only including opinions of end-users, but also of other stakeholders such as healthcare professionals or personal trainers.

We have found small differences in needs and wishes between potential users that are already active and those that are not. In further explorative analyses, we observed differences in rating based on gender and age. Future research could explore the differences in needs between different user groups in more detail in order to better tailor the experience of the application to the needs of the specific target group.

It might be the case that there are no major differences between groups when it comes to how important they rate different clusters and statements concerning features in mHealth for PA promotion. On the one hand, it could be the case that the main between-group differences lie not in *what kind* of features are desired, but in *how* these features should be implemented. While this is obvious for some features (*e.g.*, 'Personalised suggestions' should take the individual into account and not be the same for every user), other features, such as video demonstration, informational texts or planning features might also require further personalisation. When planning mHealth interventions, researchers should thus also involve end users in decision-making about the delivery of specific features, such as choices of layout, tone or format.

Whereas these results stem from a healthy adult sample, they can also inform interventions developed for clinical samples. While involving the specific group of end-users will always be an important part of intervention development, the statements or even clusters created in this study can be used as a starting point for discussions about desired and undesired features. Future GCM studies on mHealth interventions for physical activity may use our statements as a starting point, adding further statements based on specific requirements in their target group. Differences in importance ratings based on gender, age and activity level, but also other relevant factors such as severity of symptoms, will have to be taken into account.

Moreover, differences within groups might be just as or even more relevant than between group differences. Mhealth applications in both healthy and clinical samples can take these differences into account by (1) making some features optional, leaving the agency with the user, and (2) personalizing the application based on the behaviour of the user.

## STRENGTHS AND LIMITATIONS

This study used a mixed methods approach in order to map the wishes and needs of potential users of an mHealth intervention promoting physical activities. This approach unites some of the key advantages of qualitative research with the strengths of quantitative research.

When it comes to the qualitative aspects in generating statements, this study used a group brainstorming session, providing the researchers the opportunity to introduce the topic of PA promotion through mHealth. It also allowed participants to formulate statements throughout the individual brainstorming and group discussion, which might have resulted

in a broader range of statements. Moreover, brainstorming sessions provided participants with an opportunity to elaborate on their statements, providing a deeper understanding of each statement by the researchers and reducing the risk of misunderstandings.

When it comes to the quantitative aspects in rating and sorting statements, this study used an online survey, which allowed the study to be accessible to a wider range of participants, and to collect information from more participants overall.

This is one of the first studies using GCM in developing interventions for PA promotion. All statements and ideas came from the participants, and were rated by a second group of participants, with the researchers only facilitating the different forms of input and processing the information. As such, this method was user-centred and attempted to avoid steering participants in specific directions based on theory or previous research results.

This study has some limitations. First, we used a direct prompt to facilitate brainstorming. This could have led to a narrow generation of ideas. Second, all interaction with the target group were online due to the COVID-19 measures at the time of data collection. This may have excluded potential participants with less digital literacy from contributing to this project. Third, the participants that were willing to participate in this study may differ from the general population, as they showed sufficient motivation and interest in the topic of PA to participate in the study. It is likely that this affects the generalizability of our results.

## CONCLUSIONS

The current study made use of the GCM procedure to investigate the needs and wishes of potential users of an mHealth applications that aims to promote PA in the general population. The resulting cluster structure contained six clusters, namely 'Ease-of-use and Self-monitoring', 'Technical Aspects and Advertisement', 'Personalized Information and Support', 'Motivational Aspects', 'Goal Setting, Goal Review and Rewards', and 'Social Features'. Users rated 'Ease-of-use and Self-monitoring' highest, shortly followed by 'Technical Aspects and Advertisement'. Particularly low ratings was given to 'Social Aspects' of the application. Only small differences were observed between participants that had already been leading an active lifestyle and those who had not.

The present study corroborates the need for mHealth applications that aim to promote PA to be user-friendly, tailored to the individual, contain self-monitoring features and provide participants with agency, consistent with previous qualitative and quantitative research. Social features, while highly recommended by some development approaches, have been rated as unimportant by the participants of the present study. If implemented, users indicated they want to be given the choice whether they want to engage with social aspects of an mHealth intervention. Our results underline the importance of personalization beyond the group level. Whereas differences between groups were relatively small, personal preferences differed throughout all groups. Interventions can take this into account by giving agency to the user and personalizing features to each individual.

Our results of this research will inform the development of an mHealth intervention. Future research can expand this approach to other target groups, such as patient groups, or to draw comparisons between groups. An interesting addition could also be to add a healthcare provider perspective on top of a user perspective exclusively.

## ACKNOWLEDGEMENTS

We would like to thank the members of the Ghent Health Psychology Lab and the eBehaviourChange Team for their feedback on the design of this study.

### Funding

This work was funded by an interdisciplinary research grant (01IO0320) from the Special Research Fund of Ghent University. The funders had no role in study design, data collection and analysis, decision to publish, or preparation of the manuscript.

### Grant Disclosures

The following grant information was disclosed by the authors:
Special Research Fund of Ghent University: 01IO0320.

### Competing Interests

The authors declare there are no competing interests.

### Author Contributions

- Maya Braun conceived and designed the experiments, performed the experiments, analyzed the data, prepared figures and/or tables, authored or reviewed drafts of the article, and approved the final draft.
- Stéphanie Carlier conceived and designed the experiments, performed the experiments, authored or reviewed drafts of the article, and approved the final draft.
- Femke De Backere conceived and designed the experiments, authored or reviewed drafts of the article, and approved the final draft.
- Marie Van De Velde conceived and designed the experiments, performed the experiments, authored or reviewed drafts of the article, and approved the final draft.
- Filip De Turck conceived and designed the experiments, authored or reviewed drafts of the article, and approved the final draft.
- Geert Crombez conceived and designed the experiments, authored or reviewed drafts of the article, and approved the final draft.
- Annick L. De Paepe conceived and designed the experiments, authored or reviewed drafts of the article, and approved the final draft.

### Human Ethics

The following information was supplied relating to ethical approvals (*i.e.*, approving body and any reference numbers):

The Faculty of Psychology and Educational Sciences of Ghent University granted Ethical approval to carry out the study (Ethical Application Ref.: 2021-198)

### Data Availability

The data is available at open science framework: Braun, Maya, Femke De Backere, Filip De Turck, Stéphanie Carlier, Marie Van de Velde, Geert Crombez, and Annick De Paepe. 2024. ''Raw Data.'' OSF. February 16. doi:10.17605/OSF.IO/KG4VC.

## Supplemental Information

Supplemental information for this article can be found online at http://dx.doi.org/10.7717/peerj.17100#supplemental-information.

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
