# Peer review of "Identifying app components that promote physical activity: a group concept mapping study"

_PeerJ, doi:10.7717/peerj.17100_

## Round 0.1 · original submission · Minor Revisions

Thank you for your submission. The reviewers have identified a number of concerns that must be addressed.

Reviewer 1 ·

Basic reporting

- Lines 145-165 describe the advantages and the disadvantages of your group brainstorming approach and should be moved to the background or the discussion section.
- In line 280, please indicate what t-tests were used. For example, independent two-sample t-tests. Please indicate what P-value was used to determine statistical significance. For example, two-sided p-values < 0.05 were used to determine statistical significance.
- For Figure 2, please reformat the x-axis to show “0”, “1”, “2” .. “25”, with an interval of one instead of five.
- Please convert Figure 3 to a bar chart, with x-axis representing the type of cluster and y-axis showing the average importance ratings.
- Please update Figure 4. For example, in Figure (a), create 6 panels, each representing a different cluster: “technical aspects and advertisement“, “ease-of-use and self-monitoring”, “Personalized information and support”, “motivational aspects”, “goal setting, goal review and rewards”, and “social features“. In each panel, show a bar chart with x-axis indicating two groups: active subgroup and inactive subgroup, and y-axis indicating the mean ratings of that cluster. Please also add the p-values to the graph.
- Please add p-values from comparing between active vs. inactive subgroups to Table 1.
- In table 2, you don’t have to show t and df. Please show the mean and the SD in each group instead. Please also indicate if the p-values here were Bonferroni corrected or not. Please also describe how you conducted the Bonferroni correction in the statistical analysis section.

Experimental design

- In the Procedure section, please elaborate on how the statements from participants were recorded. Were they audio-recorded or recorded on digital post-its? In the data analysis section, please clarify when the statements were processed by the researchers. Were they processed during the group brainstorming? How the disagreements between the researchers were resolved? In what format the statements were accessed by the researchers who did not attend the brainstorming session?

Validity of the findings

no comment

Reviewer 2 ·

Basic reporting

(1)
The overall writing is clear, but some sentences are long and complex. Consider breaking them down for better readability.

(2)
While well-integrated with existing literature, consider citing more recent studies where applicable.

(3)
Please ensure consistent use of terminology (e.g., 'e-Health' and 'mHealth'). Clarify and use one term consistently throughout.

(4)
Please review the sentence structure and grammar for enhanced readability.

Experimental design

(1)
Good use of a mixed-methods approach, combining qualitative and quantitative methods.

(2)
The user-centred approach is appreciated, where participants generated ideas and rated them without steering based on theory.

(3)
The online interaction due to COVID-19 measures is understood. However, please consider it may have excluded those with less digital literacy, impacting generalizability.

Validity of the findings

(1)
Insightful discussion on social features. You may explore further why these features received lower ratings despite recommendations in the literature.

(2)
The discussion on credibility support is brief. You may consider expanding on why participants did not prioritize it and discuss potential implications for future interventions.

(3)
It would be beneficial to elaborate on the implications for future research. Specify aspects to explore in more detail and how findings could apply to different target groups or contexts.

(4)
The conclusions effectively summarize key findings. However, consider highlighting any novel contributions and how the study advances the field.

---

## Round 0.2 · accepted · Accept

Thank you for your revised submission. I am satisfied that you addressed the concerns of the reviewers, and am happy to accept your paper for publication.

Reviewer 1 ·

Basic reporting

The authors have adequately addressed my comments. Therefore, I have no further comments.

Experimental design

The authors have adequately addressed my comments. Therefore, I have no further comments.

Validity of the findings

The authors have adequately addressed my comments. Therefore, I have no further comments.

Reviewer 2 ·

Basic reporting

Thanks for accommodating the feedback, I have no further comments.

Experimental design

No further comments.

Validity of the findings

No further comments.